# Factors Associated with the Antibiotic Treatment of Children Hospitalized for COVID-19 during the Lockdown in Serbia

**DOI:** 10.3390/ijerph192315590

**Published:** 2022-11-24

**Authors:** Andreja Prijić, Tatjana Gazibara, Sergej Prijić, Stefan Mandić-Rajčević, Nataša Maksimović

**Affiliations:** 1Children’s Hospital for Lung Diseases and Tuberculosis, University Clinical Hospital Center “Dr Dragiša Mišović–Dedinje”, 11000 Belgrade, Serbia; 2Institute of Epidemiology, Faculty of Medicine, University of Belgrade, 11000 Belgrade, Serbia; 3Mother and Child Health Institute of Serbia “Dr Vukan Čupić”, 11000 Belgrade, Serbia; 4Institute of Social Medicine, Faculty of Medicine, University of Belgrade, 11000 Belgrade, Serbia

**Keywords:** COVID-19, children, hospital, antibiotics

## Abstract

Unselective use of antibiotics to treat children with COVID-19 is one of the major issues during the pandemic in Serbia. Thus far, there has been no evidence about the predictors of multiple antibiotic use in the treatment of children with COVID-19. The purpose of this study was to assess the prevalence of antibiotic use, as well as to examine demographic and clinical factors associated with a greater number of antibiotics and with a longer antibiotic treatment administered to hospitalized children with COVID-19 during the lockdown in Serbia. This study included all children who were hospitalized from 6 March to 31 May 2020 at the only pediatric COVID-19 hospital, and who were confirmed to have SARS-CoV-2 infection. Demographic, clinical, and laboratory data were collected from medical records. The antibiotic treatment included the use of azithromycin, cephalosporin (ceftriaxone), ampicillin-amikacin, and hydroxychloroquine. The overall prevalence of antibiotics use in children hospitalized with COVID-19 regardless of age was 47.2% (43.3% in children aged 1–5 years and 44.4% in those aged 5–17 years). In children aged 1–5 years, not having a family member affected by COVID-19 (B = −1.38, 95% confidence interval [CI] −2.43, −0.34, *p* = 0.011), having pneumonia on chest X-ray (B = 0.81, 95%CI 0.34, 1.29, *p* = 0.002), being a boy (B = −0.65, 95%CI −1.17, −0.13, *p* = 0.018), and having higher C-reactive protein (CRP) values on admission (B = 0.12, 95%CI 0.07, 0.17, *p* = 0.001) were associated with the administration of a higher number of antibiotics. These factors, along with having fever (B = 3.20, 95%CI 1.03, 5.37, *p* = 0.006), were associated with a longer duration of antibiotic treatment in children aged 1–5 years. In children aged 5–17 years, having pharyngeal erythema (B = 1.37, 95%CI 0.61, 2.13, *p* = 0.001), fever (B = 0.43, 95%CI 0.07, 0.79, *p* = 0.018), and pneumonia on chest X-ray (B = 0.91, 95%CI 0.53, 1.29, *p* = 0.001), not having rhinorrhea (B = −1.27, 95%CI −2.47, −0.08, *p* = 0.037), being a girl (B = 0.52, 95%CI 0.08, 0.97, *p* = 0.021), and having higher CRP values on admission (B = 0.04, 95%CI 0.01, 0.06, *p* = 0.006) were associated with the administration of a higher number of antibiotics. These factors, not including the absence of rhinorrhea, were associated with a longer duration of antibiotics treatment in children aged 5–17 years. Demographic, epidemiological, clinical, and laboratory parameters were associated with the use of multiple antibiotics and a longer duration of antibiotic treatment both among children aged 1–5 years and those aged 5–17 years.

## 1. Introduction

COVID-19 is an infection caused by the SARS-CoV-2, which emerged at the end of 2019, and after more than 2 years, it continues to be a major global public health challenge. Since the beginning of the COVID-19 pandemic, the therapeutic approach to hospitalized patients has undergone several modifications and stages of refinement to provide optimum care. During the first pandemic wave in the spring of 2020, it was considered that azithromycin may be beneficial. Indeed, data showed that a combination of antibiotics and antiviral drugs administered at early stages of the infection in adults provided a safe and effective therapy for COVID-19 [1].

This was later dropped, as further evidence suggested that the widely used antibiotic azithromycin had little to no effect on the COVID-19 outcomes [2]. On the other hand, dexamethasone, tocilizumab, and a combination of monoclonal antibodies were observed to lower mortality of the hospitalized patients [3,4]. However, a critical analysis during the first waves of the COVID-19 pandemic showed that in this period, antibiotics were often provided without clear and precise evidence for their use (such as bacterial superinfections) [5,6,7]. Subsequently, this has raised numerous issues regarding the emerging antimicrobial resistance [8,9], which is relevant for the Sustainable Development Goals [10]. As such, the use and misuse of antibiotics has implications for the global population’s health and well-being.

Bearing in mind all mentioned above, assessment of antibiotic use is essential to revisit the past and current practices. In fact, a careful inquiry about how antibiotics were dispensed has led to the development of tailored algorithms for hospital and home management of COVID-19 patients in the early stages of the pandemic [11]. Such guidelines are pertinent for potential future pandemics caused by respiratory viruses. Still, in order to obtain such protocols, it is crucial to assess unselective approaches to treatment, as they may be specific to certain settings due to limited resources [12,13].

In Serbia, the COVID-19 epidemic emerged in March 2020. Following the rise in newly identified cases of COVID-19, the health care delivery was organized in such a way as to repurpose some of the hospitals throughout Serbia to accommodate patients with COVID-19 only. Because of the reports on low number of the affected children compared to adults, one hospital was repurposed for the hospitalization of children with COVID-19. The state of emergency due to the COVID-19 pandemic in Serbia was declared on 15 March 2020, and was lifted on 6 May 2020. In that period, the first wave of children with COVID-19 was admitted to the pediatric referral center. Because the local guidelines on antibiotics’ use in children with COVID-19 were lacking, the initial treatment protocol included azithromycin, cephalosporin (ceftriaxone), ampicillin-amikacin, and hydroxychloroquine for children with poor general health status and fever. This approach was based on the previous empirical experience of pediatricians with the H1N1 influenza pandemic (i.e., administration of antibiotics was at the discretion of a clinician) as well as the data taken from the literature.

Empirical antibiotic treatment has been observed to lengthen the hospital stay in children with COVID-19 [14]. This may result from having a more severe clinical presentation, which requires supplementation with oxygen and mechanical ventilation [15,16]. Still, thus far, there has been no evidence about the contributors to the use of multiple antibiotics in the treatment of children with COVID-19. Some data from Serbia suggest that most adult patients who were admitted into hospital had already received at least one antibiotic, and continued to receive it while hospitalized [17]. In fact, approximately one-third of those patients received multiple antibiotics; however, it was not clear which patients’ features were associated with multiple antibiotics use or that of a longer duration [17]. In order to bridge this gap, it is necessary to provide compelling information related to clinical, but also demographic, features of COVID-19 which have contributed to the use of multiple antibiotics in children. In light of the increasing antibiotic resistance, such information may be helpful for pediatricians, researchers and policy makers in future health crises.

The purpose of this study was to: (1) assess the prevalence of antibiotics use among children hospitalized for COVID-19 during the lockdown in Serbia; (2) examine demographic and clinical factors associated with a greater number of antibiotics administered to hospitalized children with COVID-19 during the lockdown in Serbia; (3) examine demographic and clinical factors associated with a longer antibiotic treatment administered to hospitalized children with COVID-19 during the lockdown in Serbia.

## 2. Materials and Methods

### 2.1. Study Population and Sample 

All children who were confirmed to have COVID-19 in Serbia were hospitalized at the official and only COVID-19 referral center for pediatric patients in the capital city of Belgrade (Serbia). The center had the capacity of 40 beds (including 5 beds in the pediatric intensive care unit (PICU) and 5 beds in the neonatal intensive care unit (NICU)). During this first pandemic wave in Serbia, all positive children were admitted, regardless of the severity of their clinical presentation (including ambulatory cases as well).

This cross-sectional study included all children (*n* = 127) who were hospitalized from 6 March to 31 May 2020 at this center, and who were confirmed to have SARS-CoV-2 infection using real-time polymerase-chain-reaction (PCR) of the nasopharyngeal swab. This period was considered because as of 1 June 2020, the government of the Republic of Serbia allowed mass gathering events, suggesting that measures of prevention and control of COVID-19 were becoming less strict. Children in whom SARS-CoV-2 was not isolated from nasopharyngeal swabs by PCR were excluded from the study.

The calculation of the sample size in the Raosoft calculator (http://raosoft.com/samplesize.html, accessed on 19 April 2020) was based on the population aged 0–17 years (1,200,000, according to Serbian population estimates for 2020 [18]), response distribution of 7.4% (based on the average prevalence of severe clinical forms of COVID-19 in children up to 10 years [19]), a margin of error of 5%, and a confidence level of 95%. The minimum sample size was 106 children.

### 2.2. Data Collection

Demographic, clinical, and laboratory data were collected for the purpose of this study. Demographic data, medical history, and family history were obtained from the children’s parents and from the hospital records. Epidemiological data retrieved from the parents included information on living with family members who were confirmed to be affected by COVID-19 and having contact with SARS-CoV-2 positive individuals.

Clinical data included symptoms reported by the parents and those symptoms identified at the time of hospitalization. Chest X-ray, laboratory findings, and treatment modalities were also considered. The symptoms of COVID-19 in the hospitalized children included: cough, pharyngeal erythema, body temperature (fever was classified as having body temperature ≥37.0 °C), diarrhea, vomiting, rhinorrhea, nasal congestion, tachypnea, and tachycardia. Laboratory findings included oxygen saturation (assessed by pulse oximetry), C-reactive protein (CRP), leukocytes, and thrombocytes. The levels of procalcitonin were not recorded.

Pneumonias were diagnosed in two ways: one was based on clinical signs and symptoms, laboratory findings, and chest auscultation (findings indicative of pneumonia were shortness of breath, crackling sounds, and wheezing), and the other was based on chest X-ray or chest CT scan (findings indicative of pneumonia were areas of tissue infiltration seen as increased lung density). This was relevant because during the first pandemic wave, pneumonia in the early stages of COVID-19 infection was not clearly distinguishable using clinical signs and symptoms only [20]. Superinfections were not observed during the study period, as almost all children had mild to moderate clinical forms.

The antibiotic treatment during the study period included only the use of azithromycin, cephalosporin (ceftriaxone), ampicillin-amikacin, and hydroxychloroquine. None of the antibiotics were excluded from the analysis. The prevalence of antibiotic use was observed per antibiotic as well as jointly. The duration of use was calculated in days. Antibiotic use longer than 10 days was considered to be prolonged treatment.

### 2.3. Statistical Analyses

We classified the children according to their age: neonates (from birth up to 4 weeks), infants (from 4 weeks up to 1 year; not including children who turned 1), younger children (1 up to 5 years; not including children who turned 5) and older children (5–17 years). This stratification was performed because of potential different clinical presentation of COVID-19 at different ages. The description of the study sample was carried out using percentages and mean values.

Due to the small sample size, we were able to run linear regression models in the subsample of children aged <5 years and 5–17 years. We performed two separate linear regression analyses, whereby in the first model, the dependent variable was the sum of antimicrobial medications received during treatment, and in the second model the dependent variable was the duration of antimicrobial treatment (measured in days). The independent variables were classified into three models, with the variable-to-subject ratio being at least 1:10. Model 1 included gender, having chronic diseases, and having family members with confirmed COVID-19. Model 2 included gender, having cough and fever on admission, and having identified pneumonia on a chest X-ray. Model 2, in the group of children aged 5–17 years, additionally included having pharyngeal erythema and rhinorrhea, because these symptoms were only present among older children. Model 3 comprised gender, level of oxygen saturation, CRP, and leukocyte and thrombocyte counts.

We have also tested the same 3 models using logistic regression when the outcomes were azithromycin use and ceftriaxone use. However, it was not possible for models to converge due to the small number of observations. For this reason, this analysis was not feasible.

Analyses were performed in SPSS 20.0. A *p*-level of <0.05 was considered statistically significant. The map of Serbia was downloaded from https://d-maps.com/carte.php?num_car=27564&lang=en, accessed on 29 May 2020, a free-of-charge platform, under the condition that the exact URL where the original map came from was provided and no more than 10 maps were used on the same occasion [21].

### 2.4. Ethical Approval

This study was approved by the Ethics Committee of the University Clinical Hospital Center “Dr Dragiša Mišović–Dedinje” (approval no. 01-493/1-1-2022). Parents provided informed consent to use their children’s data for research purposes.

## 3. Results

A total of 127 children were treated during the first wave of the epidemic at the COVID-19 referral center. The distribution of children with COVID-19, according to place of residence, is presented in Figure 1. Most children resided in the metropolitan area of the capital city of Belgrade. Children with COVID-19 from the Serbian northern Kosovo province were also treated in the pediatric referral hospital in Belgrade.

The distribution of hospital admissions according to weeks in 2020 is presented in Figure 2. The first child with COVID-19 was admitted in week 13. In the following weeks, the number of admissions gradually increased. A drop in admissions was observed in week 19. The number of admissions after this week remained relatively stable (Figure 2).

Although neonates and infants were represented in the study sample, the highest proportion of children accounted for those aged 5–17 years. The gender composition in the four age groups was not consistent. Boys were predominant in the three oldest age groups.

### 3.1. Signs, Symptoms and Therapy of COVID-19

Clinical characteristics of children according to age groups are presented in Table 1. The vast majority of children lived with family members who were confirmed to be affected by COVID-19 (Table 1). Clinically, cough and fever were the most common signs among all groups of children. The maximum body temperature observed among all children was at 40.6 °C (105.1 °F). Overall, few children had diarrhea, rhinorrhea, nasal congestions, or had previously vomited. None of the children had tachypnea (Table 2).

Oxygen saturation on admission ranged from 95% to 99%. The range of CRP on admission was from 1.0 to 58.0. Pneumonia was seen on chest X-rays in around one fifth of children aged 1–5 and 5–17 years (Table 2).

One half of infants and approximately two thirds of older children did not receive any antimicrobial medication. Thus, the overall prevalence of antibiotic use among children hospitalized with COVID-19 during lockdown in Serbia, regardless of age, was (60/127) 47.2% (the prevalence in children aged 1–5 years was 43.3% and in children aged 5–17 years it was 44.4%). The children received azithromycin, cephalosporin (ceftriaxone), ampicillin-amikacin, and chloroquine. Most children received azithromycin and cephalosporin. Of children aged 1–5 years, 13.3% received two antibiotics. Of children aged 5–17 years, 16.0% received two antibiotics and 6.2% received 3 antimicrobial medications. Two children received supplemental oxygen. None of the children required mechanical ventilation (Table 3).

### 3.2. Linear Regression Results

The results of the linear regression models are shown in Table 4 and Table 5. Linear regression models in the sample of children aged 1–5 years showed that not having a family member affected by COVID-19, having pneumonia on chest X-ray, being boy, and having higher CRP values on admission were associated with a higher number of antimicrobial medications received during treatment. All previously mentioned characteristics and having fever were associated with a longer duration of antimicrobial treatment in children aged 1–5 years (Table 4).

Having pharyngeal erythema, fever, and pneumonia on chest X-ray, not having rhinorrhea, being girl, and having a higher CRP on admission were associated with a higher number of antimicrobial medications received during hospital treatment. All previously mentioned characteristics, not including the absence of rhinorrhea, were associated with a longer duration of antimicrobial treatment in children aged 5–17 years (Table 5).

## 4. Discussion

This study sought to identify factors associated with the use of multiple antibiotics and the longer duration of treatment of hospitalized children with antibiotics due to COVID-19. Despite COVID-19 being a viral infection, as per a review paper by Wang et al. [22], antibiotics were used in various settings to treat COVID-19 in childhood. The treatment strategy in the pediatric COVID-19 hospital in Serbia was no exception. From the analyzed data, it was identified that epidemiological as well as clinical and laboratory characteristics were associated with a longer duration of antibiotic treatment and an increased number of antibiotics administered to children with COVID-19. Factors associated with the study outcomes were largely overlapping between younger and older children, suggesting that similar characteristics may predispose children of different ages in this population to receive a more intense antibiotic treatment.

Of the epidemiological characteristics, the regression model identified that not having family members with a COVID-19 infection was a risk factor for a longer duration of antibiotic treatment, among both younger and older children. This result was surprising, because it was expected that children would catch the SARS-CoV-2 in the household via droplets. Household members may be exposed to a high concentration of SARS-CoV-2 because of prolonged contact with family. The months of March and April, in the northern hemisphere, may still be quite cold. It is, therefore, expected that ventilation of rooms using windows would not yet be adequate. Previous data suggest that SARS-CoV-2 is frequently transmitted in the household [23]. It is believed that children have a lower potential for infection transmission due to the low expression of the ACE-2 gene in the nasal epithelium [24]. This may be explained by the notion that children tend to have asymptomatic forms of COVID-19, and are unlikely to develop clinical forms that require hospital attention [25].

A demographic characteristic found by this study was that boys were more likely to receive multiple antibiotics, while girls were more likely to receive antibiotics for a longer period of time. Data on the male gender and the propensity toward COVID-19 in childhood are inconsistent. Evidence from the early stages of the COVID-19 pandemic from Wuhan suggests that males were more likely to contract the infection [26]. Some studies have suggested that boys were more likely to have a fatal COVID-19 infection [27], while others found that girls were hospitalized for longer than boys [28]. Being male was observed as a risk factor for increased mortality among adults as well [29]. This discrepancy may be explained by genetic and biological differences between genders [30] that subsequently generate gender-specific immune responses [31]. For example, men and women exhibit distinctive patterns of inflammatory cytokine secretion, whereas men who have more severe forms of COVID-19 produce higher levels of antibodies [31]. However, it is believed that testosterone plays a major role in the inhibition of inflammation, while estrogen is known to activate polymorphonuclear cells [31]. Additional focus was placed on the set of genes on the X chromosome, because men have one set of X-associated genes as opposed to two sets of genes in women, which could also modulate the immune response in COVID-19 [31]. As a result, these differences may explain a more severe clinical presentation of COVID-19 in boys and subsequent administration of multiple antibiotics, as observed in our study. 

Of the clinical symptoms, a compelling predictor of receiving multiple antibiotics and receiving a longer antibiotic treatment was having pneumonia and fever, which typically occur together. However, multiple indications for antibiotic use as an anti-inflammatory agent, prophylaxis for superinfections, or treatment for atypical pneumonia make it difficult to discern clear factors for using specific antibiotics. Another laboratory finding, elevated CRP, supported these clinical findings in that a higher level of CRP was associated with the administration of multiple antibiotics for a longer period of time. However, despite the severity of the children’s health conditions, somewhat less than one half of our pediatric patients received antibiotics. This was observed by other authors as well [8]. Nevertheless, across multiple studies of COVID-19 in children, the range of antibiotics use ranged from 19.4% to 100.0% [23].

At this early stage of the COVID-19 pandemic, the most commonly used antibiotics in our patients were azithromycin and ceftriaxone. In fact, at the beginning of the pandemic, azithromycin at a dose of 10 mg/kg (first day) and 5 mg/kg (four days) was recommended for children aged 5–18 years [32]. Evidence suggests that azithromycin exhibits a potent immunomodulatory effect and reduces inflammation and viral replication [32]. For this reason, children at our hospital were primarily treated with azithromycin. Early in the COVID-19 pandemic, oral ceftriaxone and amoxicillin were also recommended by the Italian Pediatric Society for children who had productive cough and fever, but did not have other known risk factors for severe clinical forms of COVID-19 [33].

Nowadays, recommendations for hospital treatment of COVID-19 have been revised, and antibiotics are seldom used. Specifically, in our hospital, antibiotics are not being administered for at least 5 days of the hospital stay, and the decision to administer them is based on the child’s health status and the propensity to develop secondary bacterial pneumonia. Neonates, on the other hand, are observed for 3 days after, and if recovered, they are discharged. Although no census regarding therapeutic management of COVID-19 in hospitalized children is available due to the lack of randomized controlled trials, pediatric authorities suggest that the effects of some antiviral drugs—(remdesivir), corticosteroids (dexametasone), janus kinase inhibitors (baricitinib, tofacitinib), and IL-6 receptor blockers (tocilizumab)—can be extrapolated to older children (>5 years) [34]. However, most of these medications are not being administered to hospitalized children in Serbia.

Looking at the issue from a global perspective, few patients with COVID-19 truly require antibiotic treatment [35]. However, compelling evidence suggests that antibiotics were heavily used in the hospital treatment of COVID-19 worldwide, while co-occurrence of bacterial pneumonia was estimated at 8.6% [36]. This means that numerous antibiotics were misused and, therefore, have accelerated the rise of antibiotics resistance (ABR). The ABR has wider implications for population health, because certain infections are difficult to treat or not even manageable at all. Still, in countries where the use of antibiotics is not as strictly regulated, these medications are considered as a means to treat even mild health impairments, as well as a source of relief and protection [37]. In Serbia, at the beginning of the pandemic, due to the absence of specific protocols and uncertainty about the newly identified virus, antibiotics were excessively administered to treat people with COVID-19 [38]. Over the course of the pandemic, the issues with overuse of antibiotics were acknowledged and revised in order to prevent ABR. Experiences from this pandemic could serve as a blueprint to develop the treatment guidelines in future health crises.

Previous data suggest that 65% of children with COVID-19 have some type of respiratory symptoms, about one half have fever, and 38% have coughing [8], while many fewer children present gastrointestinal symptoms. We observed that cough and fever were the most common symptoms of children hospitalized with COVID-19 in our institution, which is not surprising, as they were most commonly found as the symptoms of COVID-19 pneumonia in children [39]. Outside of the hospital, however, a large-scale study of children with COVID-19 from the UK found that headache, fatigue, and anosmia were the most common symptoms [32]. However, differences in symptoms were noted in children infected with the alpha and delta variants, with the latter inducing longer and more intense symptoms [40].

With respect to the laboratory findings in children with COVID-19, studies suggest that the neutrophil count may be increased or decreased, while the lymphocyte count is usually decreased [41], which is expected in viral infections. Elevated liver enzymes (lactate dehydrogenase—LDH, aspartate aminotransferase—AST, and alanine aminotransferase—ALT) and D-dimer have been reported in severe cases of COVID-19 [42,43]. Similarly, elevated CRP and procalcitonin levels suggest a possible secondary bacterial infection and are the key indicators of inflammation, as leukocyte count is not a reliable indicator of disease severity in children. Children in this study had a wide range of CRP values, and its levels increased with age. The widest range and the highest values of CRP were observed in older children (5–17 years). Nevertheless, CRP in younger children was also indicative of receiving multiple antibiotics for a longer period of time. As a result, monitoring of CRP is advised in order to assess the trajectory of the infection in children.

Children in our hospital were admitted based on a positive PCR test, clinical presentation, laboratory analyses, and radiography/CT of the lungs. It has been suggested that COVID-19 in children has less specific features on chest X-ray compared to that in adults [44]. Most children in this study did not have pneumonia on their chest X-ray, which is in agreement with studies conducted by other authors [44]. Those children who had abnormalities on radiographic imaging often presented unilateral lung changes and ground-glass opacities [45]. In severe pneumonia, peripheral lung tissue and a lower respiratory tract are more intensely involved in the inflammatory process [46].

Overall, our results are somewhat similar to those observed in other countries with a comparative gross domestic product per capita (i.e., upper middle income countries, as per the classification of the World Bank) [15,47]. In a pooled analysis of several Latin American countries [15] about 40% of children hospitalized in the PICU had upper respiratory tract symptoms, approximately almost one-third had gastrointestinal disturbances, contrary to our patients. Similarly to our set of children, 28% of pediatric patients in Latin America underwent chest X-ray, and in one-third of those findings, were suggestive of pneumonia [15]. However, several antiviral drugs were administered and antibiotics were used in cases of COVID-19 complications [15], which contrasts with the therapeutic approach in our study. A study in Turkey observed that children hospitalized in tertiary health centers presented with cough and fever [47], which is in line with our observations as well. The Turkish pediatricians administered azithromycin to a higher proportion of older children and ceftriaxone to fewer patients [47] compared to our study. However, in addition to these antibiotics, they included amoxicillin, teicoplanin/piperacillin/tazobactam, meropenem, and oseltamivir in the therapeutic management of children hospitalized with COVID-19 [47].

This study has some limitations. We have limited this analysis to the period of lockdown in Serbia. Over the course of the COVID-19 pandemic, the Serbian government introduced only one lockdown as a measure of pandemic prevention and control. During this time, our center was the only referral center for hospital treatment of children with COVID-19. Later in the pandemic, children were admitted to other hospitals as well. For this reason, hospitalizations of children during the lockdown period were unique, and provided the basis for the treatment of children with severe COVID-19. However, this approach has certain drawbacks, as the lockdown period was relatively short. As a result, the size of the study sample is somewhat small for the purpose of obtaining reliable results for clinicians’ decision-making. This was especially pronounced when stratification according to the age of the children was performed. As a result, the analytical strategy had to include several smaller regression models in order to maintain the sufficient robustness with a satisfactory variable-to-subject ratio. We were unable to include all the collected variables, because some of them had 0 observations, such as vomiting, diarrhea, and rhinorrhea for children aged 1–5 years. Similarly, other characteristics, such as oxygen and ventilatory support, had few observations and could not be meaningfully analyzed in the regression models. It was, therefore, not possible to understand whether children who received oxygen and ventilation had more chances of receiving multiple antibiotics. Another limitation of this study may be related to the notion that these findings are dated. However, based on the protocols for COVID-19 treatment that our hospital has defined, azithromycin, cephalosporin, ampicillin, and gentamicin are still being used, albeit only for children who have prolonged fever (>5 days for infants and older children and >2 days for neonates) or evident pneumonia. Empirical results with the antibiotic treatment seen in the early days of the pandemic could have played the role in structuring the protocols for current antibiotic use. Because this analysis pertained to the cross-sectional study methodology, it may not provide evidence for causal inference. To understand more thoroughly potential long-term effects of COVID-19 infection, a prospective follow-up of this study sample is warranted.

## 5. Conclusions

In conclusion, most children who were hospitalized during the first COVID-19 pandemic wave in Serbia in 2020 did not receive antibiotic treatment. Few children received more than one antibiotic while being hospitalized. Overall, the maximum duration of antibiotic therapy was 7 days. Demographic, epidemiological, clinical, and laboratory parameters were associated with the use of multiple antibiotics and a longer duration of antibiotic treatment, both among children aged 1–5 years and those aged 5–17 years.

## Figures and Tables

**Figure 1 ijerph-19-15590-f001:**
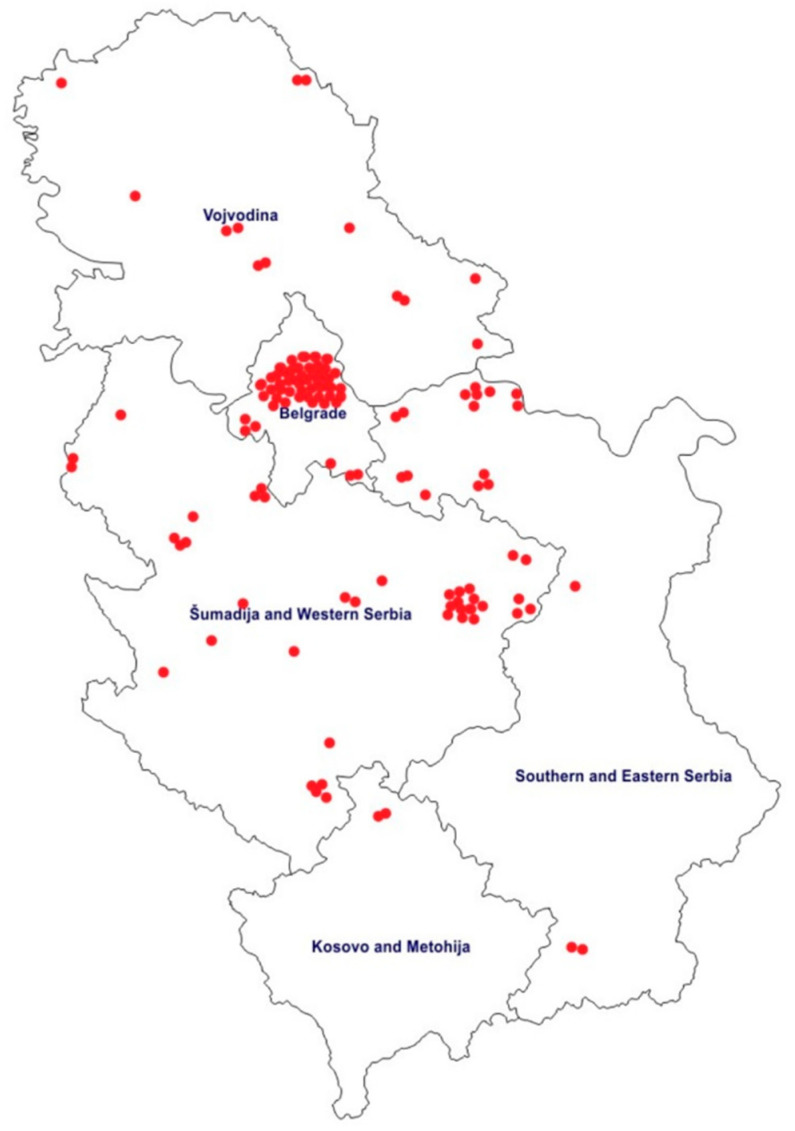
Topographic distribution of children with COVID-19 during the first pandemic wave in Serbia.

**Figure 2 ijerph-19-15590-f002:**
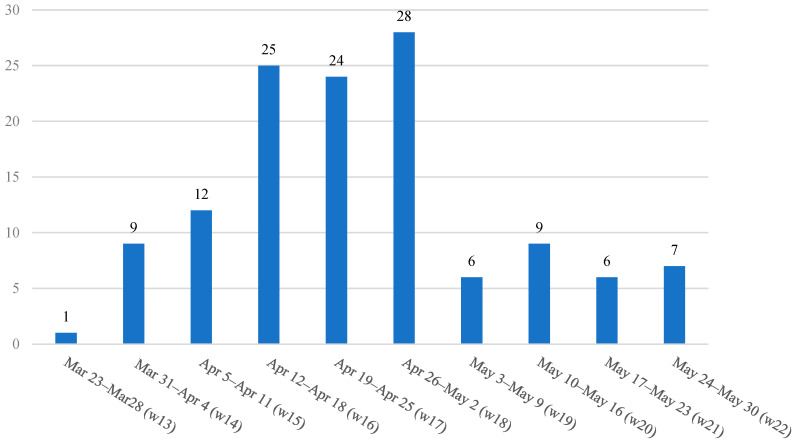
Chronological distribution of children admitted to the pediatric COVID-19 hospital in 2020 during the first pandemic wave in Serbia.

**Table 1 ijerph-19-15590-t001:** Demographic characteristics of children diagnosed with COVID-19 according to age groups (*n* = 127).

Variable	Neonates	Infants	Children 1–5 y	Children Aged 5–17 y
	*N* = 6	*N* = 10	*N* = 30	*N* = 81
	*n* (%)	*n* (%)	*n* (%)	*n* (%)
Gender				
Male	2 (33.3)	8 (80.0)	17 (56.7)	47 (58.0)
Female	4 (66.7)	2 (20.0)	13 (43.3)	34 (42.0)
Chronic diseases				
Yes	0 (0)	0 (0)	3 (10.0)	10 (12.3)
No	6 (100.0)	10 (100.0)	27 (90.0)	71 (87.7)
Confirmed COVID-19 in family				
Yes	6 (100.0)	9 (90.0)	22 (73.3)	77 (95.1)
No	0 (0)	1 (10.0)	8 (26.7)	4 (4.9)

**Table 2 ijerph-19-15590-t002:** Clinical characteristics of children diagnosed with COVID-19 according to age groups (*n* = 127).

Variable	Neonates	Infants	Children 1–5 y	Children Aged 5–17 y
	*N* = 6	*N* = 10	*N* = 30	*N* = 81
	*n* (%)	*n* (%)	*n* (%)	*n* (%)
Cough				
Yes	6 (100.0)	7 (70.0)	20 (66.7)	60 (74.1)
No	0 (0)	3 (30.0)	10 (33.3)	21 (25.9)
Pharyngeal erythema				
Yes	0 (0)	1 (10.0)	0 (0)	5 (6.2)
No	6 (100.0)	9 (90.0)	30 (100.0)	76 (93.8)
Fever				
Yes	2 (33.3)	7 (70.0)	12 (40.0)	37 (45.7)
No	4 (66.7)	3 (30.0)	18 (60.0)	44 (54.3)
Body temperature when having fever (range)	37.7 °C (37.3–38.2) 99.8 °F (99.1–100.7)	38.3 °C (37.3–40.6) 100.9 °F (99.1–105.1)	38.2 °C (37.2–39.4) 100.7 °F (99.0–102.9)	38.0 °C (37.1–39.5) 100.4 °F (98.8–103.1)
Duration of fever in days (range)	1 *	2.7 (1–6)	3.8 (1–8)	3.3 (0–10)
Diarrhea				
Yes	0 (0)	2 (20.0)	0 (0)	3 (3.7)
No	6 (100.0)	8 (80.0)	30 (100.0)	78 (96.3)
Vomiting				
Yes	1 (16.7)	2 (20.0)	0 (0)	1 (1.2)
No	5 (83.3)	8 (80.0)	30 (100.0)	80 (98.8)
Rhinorrhea				
Yes	0 (0)	1 (10.0)	0 (0)	2 (2.5)
No	6 (100.0)	9 (90.0)	30 (100.0)	79 (97.5)
Nasal congestion				
Yes	1 (16.7)	4 (40.0)	0 (0)	1 (1.2)
No	5 (83.3)	6 (60.0)	30 (100.0)	80 (98.8)
Tachypnea				
Yes	0 (0)	0 (0)	0 (0)	0 (0)
No	6 (100.0)	10 (100.0)	30 (100.0)	81 (100.0)
Tachycardia				
Yes	1 (16.7)	0 (0)	0 (0)	0 (0)
No	5 (83.3)	10 (100.0)	30 (100.0)	81 (100.0)
Oxygen saturation % (range)	98 (96–99)	97 (96–99)	98 (95–99)	98 (96–99)
C-reactive protein (range)	3.8 (1.0–5.30)	5.8 (1.0–9.50)	4.1 (1.0–25.4)	4.3 (1.0–58.0)
Leukocytes (×10^9^/L)	11.1 (8.9–12.5)	8.3 (5.0–15.0)	7.4 (4.3–13.9)	6.8 (2.8–14.9)
Thrombocytes (×10^9^/L)	277.4 (212.0–372.0)	308.1 (186.0–432.0)	309.0 (119–779)	237.8 (57.0–410.0)
Pneumonia seen on chest X-ray				
Yes	0 (0)	1 (10.0)	8 (26.7)	21 (28.0)
No	6 (100.0)	9 (90.0)	22 (73.3)	60 (72.0)

Legend: All continuous variables are presented as means (minimum–maximum); * only two neonates had fever (both had fever for 1 day).

**Table 3 ijerph-19-15590-t003:** Characteristics of therapy administered to children diagnosed with COVID-19 according to age groups (*n* = 127).

Variable	Neonates	Infants	Children 1–5 y	Children Aged 5–17 y
	*N* = 6	*N* = 10	*N* = 30	*N* = 81
	*n* (%)	*n* (%)	*n* (%)	*n* (%)
Azithromycin				
Yes	1 (16.7)	2 (20.0)	13 (43.3)	32 (39.5)
No	5 (83.3)	8 (80.0)	17 (56.7)	49 (60.5)
Cephalosporin (ceftriaxone)				
Yes	1 (16.7)	2 (20.0)	4 (13.3)	22 (27.2)
No	5 (83.3)	8 (80.0)	26 (86.7)	59 (72.8)
Ampicillin-amikacin				
Yes	1 (16.7)	1 (10.0)	0 (0)	0 (0)
No	5 (83.3)	9 (90.0)	30 (100.0)	81 (100.0)
Chloroquine				
Yes	0 (0)	0 (0)	0 (0)	5 (6.2)
No	6 (100.0)	10 (100.0)	30 (100.0)	76 (93.8)
Number of medications used				
0	0 (0)	5 (50.0)	17 (56.7)	45 (55.6)
1	5 (83.3)	5 (50.0)	9 (30.0)	18 (22.2)
2	1 (16.7)	0 (0)	4 (13.3)	12 (16.0)
3	0 (0)	0 (0)	0 (0)	5 (6.2)
Maximum duration of therapy in days for children who received medications (range)	7.0 *	5.6 (5–7)	6.4 (5–11)	6.4 (5–10)
Supplemental oxygen				
Yes	0 (0)	0 (0)	1 (3.3)	1 (1.2)
No	6 (100.0)	10 (100.0)	29 (96.7)	80 (98.8)
Mechanical ventilation				
Yes	0 (0)	0 (0)	1 (3.3)	0 (0)
No	6 (100.0)	10 (100.0)	29 (96.7)	81 (100.0)

Legend: All continuous variables are presented as means (minimum–maximum). * All neonates received therapy for 7 days.

**Table 4 ijerph-19-15590-t004:** Results of linear regression models in children aged 1–5 years: factors associated with the use of a greater number of medications and a longer duration of medication therapy.

Variable	Total Number of Medications Used			Duration of Medication Therapy		
	B	95%CI	*p*	B	95%CI	*p*
Socio-epidemiologic model						
Gender Male vs. female	−0.29	−0.78, 0.20	0.232	−2.13	−4.48, 0.23	0.075
Having chronic diseases Yes vs. no	0.04	−0.81, 0.88	0.925	0.96	−3.02, 4.95	0.623
Confirmed positive family members Yes vs. no	−1.38	−2.43, −0.34	**0.011**	−5.16	−10.08, −0.23	**0.041**
Model summary	R^2^ = 0.325, adj R^2^ = 0.248, *p* = 0.015			R^2^ = 0.340, adj R^2^ = 0.261, *p* = 0.014		
Clinical symptoms model						
Gender Male vs. female	−0.14	−0.60, 0.32	0.532	−1.76	−3.61, 0.08	0.060
Cough Yes vs. no	0.29	−0.26, 0.84	0.292	0.25	−2.12, 2.63	0.826
Fever Yes vs. no	0.44	−0.07, 0.85	0.086	3.20	1.03, 5.37	**0.006**
Pneumonia on X-ray Yes vs. no	0.81	0.34, 1.29	**0.002**	4.02	2.07, 5.97	**0.001**
Model summary	R^2^ = 0.546, adj R^2^ = 0.470, *p* = 0.001			R^2^ = 0.684, adj R^2^ = 0.629, *p* = 0.001		
Laboratory findings model						
Gender Male vs. female	−0.65	−1.17, −0.13	**0.018**	−3.48	−6.14, −0.82	**0.013**
Oxygen saturation	−0.12	−0.35, 0.10	0.253	−0.82	−1.95, 0.32	0.147
C-reactive protein	0.12	0.07, 0.17	**0.001**	0.37	0.10, 0.64	**0.011**
Leukocytes	−0.07	−0.20, 0.05	0.236	−0.04	−0.71, 0.62	0.893
Thrombocytes	0.01	−0.01, 0.01	0.210	0.01	−0.01, 0.01	0.266
Model summary	R^2^ = 0.679, adj R^2^ = 0.594, *p* = 0.001			R^2^ = 0.639, adj R^2^ = 0.539, *p* = 0.001		

Legend: B—unstandardized regression coefficient; CI—confidence interval; *p*—probability level; R^2^—coefficient of determination; adj R^2^—adjusted coefficient of determination. Bolded values represent statistical significance, *p* < 0.05.

**Table 5 ijerph-19-15590-t005:** Results of linear regression models in children aged 5–17 years: factors associated with the use of a greater number of medications and a longer duration of medication therapy.

Variable	Total Number of Medications Used			Duration of Medication Therapy		
	B	95%CI	*p*	B	95%CI	*p*
Socio-epidemiologic model						
Gender Male vs. female	0.31	−0.12, 0.74	0.158	0.99	−0.55, 2.52	0.204
Having chronic diseases Yes vs. no	0.23	−0.39, 0.85	0.460	−0.01	−2.24, 2.21	0.991
Confirmed positive family members Yes vs. no	−0.86	−1.83, 0.12	0.086	−4.05	−7.55, −0.55	**0.024**
Model summary	R^2^ = 0.092, adj R^2^ = 0.057, *p* = 0.057			R^2^ = 0.107, adj R^2^ = 0.072, *p* = 0.032		
Clinical symptoms model						
Gender Male vs. female	0.27	−0.08, 0.61	0.129	1.04	−0.16, 2.24	0.089
Cough Yes vs. no	0.07	−0.33, 0.47	0.737	−0.58	−1.97, 0.82	0.413
Pharyngeal erythema Yes vs. no	1.37	0.61, 2.13	**0.001**	3.08	0.44, 5.72	**0.023**
Fever Yes vs. no	0.43	0.07, 0.79	**0.018**	2.29	1.07, 3.52	**0.001**
Diarrhea Yes vs. no	0.92	−0.04, 1.88	0.059	2.05	−1.26, 5.36	0.221
Rhinorrhea Yes vs. no	−1.27	−2.47, −0.08	**0.037**	−3.80	−7.92, 0.32	0.070
Pneumonia on X-ray Yes vs. no	0.91	0.53, 1.29	**0.001**	3.88	2.57–5.18	**0.001**
Model summary	R^2^ = 0.502, adj R^2^ = 0.450, *p* = 0.001			R^2^ = 0.536, adj R^2^ = 0.487, *p* = 0.001		
Laboratory findings model						
Gender Male vs. female	0.52	0.08, 0.97	**0.021**	1.68	0.06, 3.30	**0.043**
Oxygen saturation	−0.18	−0.47, 0.10	0.206	−0.49	−1.53, 0.55	0.351
C-reactive protein	0.04	0.01, 0.06	**0.006**	0.14	0.05, 0.24	**0.004**
Leukocytes	−0.02	−0.12, 0.07	0.650	−0.09	−0.44, 0.25	0.584
Thrombocytes	−0.01	−0.01, 0.01	0.478	0.01	−0.01, 0.01	0.744
Model summary	R^2^ = 0.224, adj R^2^ = 0.164, *p* = 0.005			R^2^ = 0.199, adj R^2^ = 0.137, *p* = 0.012		

Legend: B—unstandardized regression coefficient; CI—confidence interval; *p*—probability level; R^2^—coefficient of determination; adj R^2^—adjusted coefficient of determination. Bolded values represent statistical significance *p* < 0.05.

## Data Availability

Dataset underlying this study is available on a reasonable request to the corresponding author.

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
