# Peer review of "Factors Associated with the Antibiotic Treatment of Children Hospitalized for COVID-19 during the Lockdown in Serbia"

_ijerph, 2022, doi:10.3390/ijerph192315590_

Round 1
Reviewer 1 Report
The submitted work is well-presented and provides valuable findings, however, there are a few aspects that should be taken into consideration before publishing in the journal of IJERPH. My minor comments and suggestions are given below:
§ There are plenty of grammatical and sentence structuring mistakes throughout the paper. The authors need another attempt to make it more valuable and well-presented for its readership.
§ The first sentence of the abstract is not a good fit. The abstract doesn’t start with the conjunction, interjection, or adverb (“because”). Please rewrite.
§ Don’t you think the data period is too old and also too short as well. Consequently, the sample size is too small to get reliable results for clinicians' decision-making. Please elaborate.
§ The following claim reported by the authors doesn’t contain any literature evidence.
“During the first pandemic wave in the spring of 2020, it was considered that azithro-58 mycin and hydroxychloroquine may be beneficial. This was later dropped, while dexa-59 methasone, tocilizumab and combination of monoclonal antibodies were observed to 60 lower mortality of the hospitalized patients”
§ As the study has been done for the Serbian population, authors must include some background and related studies done before to set the literature gaps. Why did the authors need to conduct this study? Why authors chose Serbia? Why it was a single-center study? What was done before and what was left to do the further investigation? By keeping these things in your mind, rewrite your literature gaps and then write the contribution part again at the end of the introduction.
§ The introduction part requires more recent data to build the arguments. Please add further references to support your background.
§ The discussion is well-described. The authors should add more recent references to make the discussion stronger and more impactful.
§ Under each table, please mention the acronyms used within the table and also the significant level of p values.
§ There are a few in-text citations that are not present in the references. Please revise them.
Author Response
The answers and proposed changes are available in the attached document.

Reviewer 2 Report
Dear authors. I received the article entitled: "Factors associated with antibiotic treatment of children hospitalized for COVID-19: an analysis of the first two months of the coronavirus epidemic 2020 in Serbia" for review. Despite the relevance, some important points should be reviewed.
1. Abstract- Please present numeric data of results in this section, to sustain your affirmation
2- Material and methods- Please describe bed-capacity of the hospital, describe number of beds available for children, includind different wards ( clinical wards, PICU, NICU, etc) . Please clarify if all positive cases of COVID-19, in Serbia during this period were admitted, independent of severity? (even ambulatory cases?). Please inform if only azithomicin, cephalosporin, ampicillin, amikacin and hydroxychloroquine were used. Other classes were not used? Please define how many days were considered long time of antibiotic treatment?
3- Results
Table 1 is too long. Maybe you could separate in three parts: a) Demographic data, b) Signals and symptoms and c) Medications/ length of stay. Please also inform which means the numbers in parenthesis (%) . Please review information in line 162 ( is different from table 1) . Please inform meaning of letter B and 95%CI
4- Discussion
Lines 225 to 227- Please review this sentence, using the WHO Therapeutics and COVID-19 guideline
Points that could be discussed- Few data about treatment were available during the first wave. Please compare it with the current knowledge.
Author Response

(The authors gave the same response as above.)

Reviewer 3 Report
Dear authors
A very interesting paper with an important message
I have based my review on the following questions: a) Is the title accurate and relevant?, b) Does the abstract adequately summarise the work and conclusions drawn?, c) Is the presentation good?, d) Does the English require attention?, e) Would the paper benefit from shortening?, f) Is the statistical methods used accurate and used correctly?, g) Is the work involving humans, animals or bacteria strain samples, and are the adequate ethical standards being used? h) Are all figures and tables necessary and satisfactory? And is there repetition of data in the text?, i) Are the names of organisms, chemical and drug names used appropriately? j) Are the level of references appropriate and does the author have the latest knowledge? k) Does the paper contain data that enlarge our knowledge of the subject? If new data is present but at low levels, would the paper be better presented as a letter? l) What priority should the paper be given?
hydroxychloroquine
Here are some suggestions.
Reformat rows 57-64 rewrite this text since the risk of hydroxychloroquine might be diffecult to have as a text here.
If possible add more numbers since the sample size are low for the different groups.
Suggestion to slit the one of the groups into two in the classification.Classification 1-5 years is diffecult since many studies show a change in uptake from 3 years of age and also colonization of AMR, and covid 19 pathogenes are different below 3 years and above.
Suggestion to shorten the discussion with at least one paragraph, before the paragraph about limitations.
Suggestion to add a ref and discussion about a country with similar economic situation as Serbia
a) The title is accurate but could be shorted
b) The abstract summarize fine but in it there are some statements that could be rewritten to push the message. so I suggest a rewrite.
c) The presentation in general is good but there are some issues, with risky statements that we all now know are risky behavior and the article should avoid these.. statements
d) The English is very good and with so many european speaking authors I would have been surprise otherwise.
e) I think the article length is some what well used and could even be a little longer with a comparison with a country with same political or economic situation as Serbia.
f) Yes methods are accurate but usage of statistical programs might have shown uni or multivariable results that would benefit the article.
g) The ethical permit /approval is correct.
h) No the figures and tables are accurate
i) YES to my understanding yes.
j) well, the authors should be aware of other more recent publications and maybe some shift of a more global perspective would be good.
k) The article adds only a few new points to the global knowledge but more to the local one, therefor I suggest some new global comparison or pushing what is new before discussion that could be shortened.
l) Rewrite and then high priority
Author Response

(The authors gave the same response as above.)

Reviewer 4 Report
This research presents risk factors for antibiotic use. Major comments: 1. Need to mention that antibiotic prescribing was at the discrection of the clinician. 2. Need to mention if any local guidelines on antibacterial use in COVID-19 children were available during the study period. I'm assuming that none were available. 3. Need to distinguish diagnosis of pneumonia from xray findings of pneumonia.(line 96). This is a critical aspect of the paper as antibiotic use will be influenced by xray findings. Need to mention what findings on xray were used to define pnuemonia. 4. Need to mention a list of cephalosporins that were used (line 64 and throughout text).(line 98) Also need to specify if any antibiotics were excluded from analysis. 5. Need to mention if superfinfections was excluded in the study, and the incidence of superfinctions. (line 82). Need to mention if procalcitonin was available or not as a test (line 81). In the discussion, suggest to explain clearly how gender may impact antibiotic use. For example, could discuss if male gender having more severe desease would generate more antibiotic use. Line 212, need to distinguish between diagnosis of bacterial pneumonia from the xray findings of pneumonia. This will be confusing for the reader. Line 232, explain clearly that the multiple indications of antibiotics as anti-inflammatory agents, prophylaxis for superinfections or treatment for atypical bacterial coverage make it difficult to discern clear factors for using specific antibiotics. Could perform a subanalysis for azithromycin only. If numbers are too low, then would mention that a specific analysis per antibiotic was not feasible. Need to rewrite entire conclusion. The english level of the conclusion can be raised to match the level of the rest of the paper. Again, need to distinguish pneumonia defined clinically from xray findings. I advise against saying that viral infections required antibacterial treatment. The last sentence of the conclusion has nothing to do with this study. Please conclude on what the results of this study are presenting.
Minor comments: In figure 2, insert the year of data being 2020 in the legend or line headings. Line 158: typo 'infant' instead of 'infants'. Line 228 replace 'per os' by 'oral'. Line 243, typo 'neutrophil' instead of 'neutrophils'
Author Response

(The authors gave the same response as above.)

Round 2
Reviewer 2 Report
Dear authors. All queries were answered
Author Response
Dear Reviewer,
Thank you for reviewing our Manuscript again.
We quote "Dear authors. All queries were answered".